# The Anti-Inflammatory and Antioxidant Properties of n-3 PUFAs: Their Role in Cardiovascular Protection

**DOI:** 10.3390/biomedicines8090306

**Published:** 2020-08-25

**Authors:** Francesca Oppedisano, Roberta Macrì, Micaela Gliozzi, Vincenzo Musolino, Cristina Carresi, Jessica Maiuolo, Francesca Bosco, Saverio Nucera, Maria Caterina Zito, Lorenza Guarnieri, Federica Scarano, Caterina Nicita, Anna Rita Coppoletta, Stefano Ruga, Miriam Scicchitano, Rocco Mollace, Ernesto Palma, Vincenzo Mollace

**Affiliations:** 1Institute of Research for Food Safety and Health (IRC-FSH), Department of Health Sciences, University “Magna Graecia” of Catanzaro, 88100 Catanzaro, Italy; oppedisanof@libero.it (F.O.); robertamacri85@gmail.com (R.M.); gliozzi@unicz.it (M.G.); v.musolino@unicz.it (V.M.); carresi@unicz.it (C.C.); jessicamaiuolo@virgilio.it (J.M.); boscofrancescabf@libero.it (F.B.); saverio.nucera@hotmail.it (S.N.); mariacaterina.zito@libero.it (M.C.Z.); lorenzacz808@gmail.com (L.G.); federicascar87@gmail.com (F.S.); caterina.nicita@gmail.com (C.N.); annarita.coppoletta@libero.it (A.R.C.); rugast1@gmail.com (S.R.); miriam.scicchitano@hotmail.it (M.S.); rocco.mollace@gmail.com (R.M.); palma@unicz.it (E.P.); 2Division of Cardiology, University Hospital Policlinico Tor Vergata, 00133 Rome, Italy; 3IRCCS San Raffaele Pisana, 00163 Roma, Italy

**Keywords:** n-3 PUFAs, oxidative stress, endogenous antioxidants, anti-inflammatory response, cardiovascular diseases

## Abstract

Polyunsaturated fatty acids (n-3 PUFAs) are long-chain polyunsaturated fatty acids with 18, 20 or 22 carbon atoms, which have been found able to counteract cardiovascular diseases. Eicosapentaenoic acid (EPA) and docosahexaenoic acid (DHA), in particular, have been found to produce both vaso- and cardio-protective response via modulation of membrane phospholipids thereby improving cardiac mitochondrial functions and energy production. However, antioxidant properties of n-3 PUFAs, along with their anti-inflammatory effect in both blood vessels and cardiac cells, seem to exert beneficial effects in cardiovascular impairment. In fact, dietary supplementation with n-3 PUFAs has been demonstrated to reduce oxidative stress-related mitochondrial dysfunction and endothelial cell apoptosis, an effect occurring via an increased activity of endogenous antioxidant enzymes. On the other hand, n-3 PUFAs have been shown to counteract the release of pro-inflammatory cytokines in both vascular tissues and in the myocardium, thereby restoring vascular reactivity and myocardial performance. Here we summarize the molecular mechanisms underlying the anti-oxidant and anti-inflammatory effect of n-3 PUFAs in vascular and cardiac tissues and their implication in the prevention and treatment of cardiovascular disease.

## 1. Introduction

Nutraceutical supplementation is considered a viable approach for prophylaxis and treatment of heart diseases [1,2,3,4]. In particular, the efficacy of polyunsaturated fatty acids (PUFAs) in the treatment of various pathologies and, especially, of cardiovascular diseases is known since long time and a large body of clinical data on their protective effect has been collected over the last decades [5,6,7,8], though several pathophysiological implications on their use still remain unclear [9].

Humans and other mammals are not able to synthesize [10]. Therefore, linoleic acid (LA, 18:2, omega-6) and α-linolenic acid (ALA, 18:3, omega-3), denoted as essential fatty acids, must be included in the diet. Docosahexaenoic acid (DHA, 22:6 n-3) and eicosapentaenoic acid (EPA, 20:5 n-3) are members of n-3 PUFAs subfamily of PUFAs being obtained from the precursor ALA or dietary fish oils; ALA, in turn, can also be found in nuts and leafy vegetables. For the prevention of cardiovascular disease (CVD), a higher consumption of n-3 PUFAs is recommended. Indeed, many studies have shown that an increase in consumption of n-3 PUFAs leads to a decrease in cases of cardiovascular disease. In particular, it has been reported that eating fatty fish a few times a week halves coronary heart disease (CHD) deaths and reduces the probability of death from a heart attack by one third [11,12,13,14]. The beneficial effects of n-3 PUFAs are due to a set of various mechanisms of action. Indeed, they have an antiarrhythmic and antithrombotic action, reduce plasma triglyceride levels and resolve inflammatory states, regulate the expression of several genes and transcription factors, and act on membrane fluidity [15,16]. In many pathologies, free radicals (such as peroxynitrite) increase proinflammatory cytokines synthesis, regulate the cyclooxygenase (COX) pathway, and promote proinflammatory prostaglandin E2 (PGE_2_) synthesis [17]. In the treatment of these diseases, n-3 PUFAs fatty acids (ω-3) have the same effects as many antioxidants, in fact, they protect endothelial cells and cardiomyocytes from damage and cell death [18]. In addition to the collected evidence, demonstrating the beneficial responses of dietary n-3 PUFAs supplementation on the cardiovascular system, some concerns have been expressed about their potential detrimental effects in different tissues due to their potential pro-oxidant effect [19]. In particular, oxidative stress and n-3 PUFAs are strongly correlated. Long- or very long-chain fatty acids have many “fragile” double bonds between carbon atoms. Therefore, the n-3 PUFAs undergo lipid peroxidation during oxidative stress, generating highly cytotoxic reactive products [20]. This is particularly dangerous in the central nervous system (CNS) [21]. In particular, it is not clear whether or not exogenously given PUFAs have a safety profile which would allow for their extensive use in the general population. Based on these concerns, the European Food Safety Authority (EFSA) expressed its opinion and claimed that, for any population group, there is not enough data to establish a tolerable upper intake level for n-3 LC PUFA (DHA, EPA and Docosapentaenoic acid (DPA), individually or combined). Indeed, for the adult population, the supplemental intakes of EPA and DHA combined at doses up to 5 g/day, and additional intakes of EPA alone up to 1.8 g/day, do not raise safety concerns. Furthermore, for the general population, the supplemental intakes of DHA alone up to 1 g/day do not raise safety concerns [22]. 

In this review, we summarize some of the recent advances regarding the potential antioxidant effects of n-3 PUFAs supplementation on the cardiovascular system along with their anti-inflammatory and cardioprotective effect in vitro and in vivo.

## 2. Antioxidant and Antinflammatory Properties of n-3 PUFAs

### 2.1. Mitochondrial Oxidative Stress and n-3 PUFAs

Molecular and cellular research shows that n-3 PUFAs have cardioprotective effects [23]. A diet rich in n-3 PUFAs has a beneficial effect on blood pressure, heart rate, left ventricular diastolic filling, and endothelial functions [24]. Therefore, n-3 PUFAs have a beneficial effect on cardiac hemodynamic factors [25]. In particular, PUFAs act as antioxidants when they are inserted into the cell membranes and regulate the antioxidant signalling pathways. Mitochondrial membranes of eukaryotic cells have a high DHA content indicating that DHA is a fundamental phospholipid for adenosine triphosphate (ATP) synthesis by oxidative phosphorylation. In mitochondria, DHA acts on many pathways; reducing oxidative stress and cytochrome c oxidase (complex IV) activity as well as increasing manganese-dependent superoxide dismutase (Mn-SOD) activity. Rats fed a high fish oil diet demonstrate a higher expression and activity of the antioxidant enzyme superoxide dismutase (SOD) and an inhibition of membrane peroxidation process as expressed by the reduced content of thio-barbituric acid (TBARS) products. The action of oxidized n-3 PUFAs is directed against Kelch-like ECH-associated protein 1 (Keap1), the negative regulator of the nuclear factor erythroid 2–related factor 2 (Nrf2). Dissociation of Keap1 from Cullin3 induces Nrf2-dependent antioxidant gene expression [26]. Additional antioxidant action of n-3 PUFAs reduces myocytes sensitivity to reactive oxygen species (ROS)-induced ischemia reperfusion (I/R) injury and increases SOD and glutathione peroxidase (GSH-Px) levels. Hypoxia condition decreases oxidative markers and increases antioxidant enzymes expression, whereas oxidized n-3 PUFAs act on sirtuin1 (SIRT1) and forkhead box (FOXO) protein, thereby increasing SOD levels. Furthermore, in human and rat studies, EPA and DHA determined F2-isoprostane reduction, an oxidative stress marker in urine [27,28].

Thus, n-3 PUFAs supplementation leads to substantial antioxidant response, an effect that occurs mainly via restoring imbalanced endogenous antioxidant moieties.

### 2.2. Nitric Oxide, Endothelial Dysfunction and n-3 PUFAs

An effect of n-3 PUFA fatty acids, which correlates with their antioxidant properties, is represented by counteraction of endothelial dysfunction, an effect which reduces arterial stiffness [29,30]. Endothelial dysfunction is determined by a reduction in endothelium-dependent vasodilation due to impaired nitric oxide (NO) bioavailability [31,32]. In fact, NO is the most important molecule for vasodilation, derived from the endothelium. In addition, NO has an anti-atherosclerotic action, reducing smooth muscle cell proliferation, platelet aggregation, and leukocyte adhesion [33]. It is known that an increase in oxidative stress induces endothelial dysfunction since ROS reduce NO bioavailability and increase synthesis of the most toxic species, i.e., peroxynitrite [34]. Changes in NO bioavailability cause procoagulable and prothrombotic conditions and vascular inflammation [35]. Important receptor-mediated cell signal transduction pathways, including the NO-cGMP pathway, are present in caveolae and lipid rafts of endothelial cell membranes [36]. n-3 PUFAs may regulate the caveolae composition resulting in increased NO production [37]. In particular, n-3 PUFA fatty acids stimulate endothelial nitric oxide synthase (eNOS) activity and expression [38]. Moreover, EPA determines a higher NO synthesis by increasing AMP-activated protein kinase (AMPK)-induced eNOS activation and eNOS dissociation from inhibitory scaffolding protein caveolin [39] (Figure 1). eNOS activity is also stimulated by DHA which favours the binding between eNOS and heat shock protein 90 (HSP-90), with activation of the PKB/AKt pathway. Thus, eNOS is phosphorylated and activated [18,40,41] and this represents the most relevant consequence of antioxidant-response elicited by supplementation with n-3 PUFAs.

### 2.3. Cell Membranes and Anti-Inflammatory Effects of n-3 PUFAs

Modulation of the cell membrane properties represents a supplemental molecular mechanism by which n-3 PUFAs lead to cardio-protective and vaso-protective response. In fact, linoleic acid and α-linolenic acid are fundamental constituents of cell membranes and are able to determine an impaired fluidity of membrane. Therefore, they determine and affect the behaviour of membrane-bound enzymes and receptors [42,43,44]. In particular, evidence has been collected showing that n-3 PUFAs inhibit both the interleukin (IL)-1, 2, 6 synthesis and the protein kinase C signalling pathway [45]. This correlates with studies showing that n-3 PUFAs interfere in various ways in inflammatory processes, inhibiting IkB phosphorylation and therefore the NF-kB signaling pathway, or through PPARα/γ, or through a ligand for GPR120, which reduces both TLR-4 and tumour necrosis factor alpha (TNF-α) signalling pathway [42,43]. Moreover, n-3 PUFAs regulate leukotrienes, prostaglandins and thromboxanes synthesis, through activation of cytosolic phospholipase A_2_ (cPLA_2_), cyclooxygenase 2, and production of PGE_2_, as a cPLA_2_ inhibitor, modify the metabolic pathway of arachidonic acid [17,18,46,47].

### 2.4. N-3 PUFA-Derived Mediators

Maresins, protectins, and resolvins are called specialized pro-resolving mediators (SPMs). They derive from ω-3 and have anti-inflammatory properties [48]. 

It is known that atherosclerosis is an unresolved inflammatory pathology in which vascular wall damage occurs. This damage is likely to be caused by reduced synthesis of SPMs, therefore it is probable that an increase in SPMs could reduce the local inflammatory response and resolve the damage caused by atherosclerosis [49]. Numerous in vivo studies attest to the cardioprotective role of specialized pro-resolving mediators. Resolvin E series, such as RvE1, are synthesized from eicosapentaenoic acid and resolve acute inflammation states as they switch off leukocyte trafficking, favour the clearance of inflammatory cells and debris as well as inhibit cytokine synthesis [50,51,52,53]. In rats treated with RvE1, after I/R, myocardial infarct size was reduced by 70% compared to control group, when administered before reperfusion. In particular, the results demonstrate that RvE1 reduces heart damage by direct action on cardiomyocytes [54].

On the contrary, biosynthesis of D-series resolvin, maresins, and protectins starts with DHA. For example, PD1 protects against brain ischemia and renal I/R injury while Resolvin D1 (RvD1) is effective against I/R and atherosclerosis [55,56].

Even after an infarct, RvD1 administration resolves the acute inflammation as it stimulates the specialized pro-resolving mediator synthesis in the spleen and favours their transfer to the anti-inflammatory M2 macrophages in the left ventricle, preventing the onset of cardiac fibrosis and ensuring the normal heart function. Epoxides, such as epoxyeicosatetraenoic acids (EpETEs) and epoxydocosapentaenoic acids (EpDPAs), are obtained from EPA and DHA with reactions catalyzed by CYP450 monooxygenase and are termed lipid mediators. They perform an anti-inflammatory action in cardiovascular disease, dilate the pulmonary arteries, activate the smooth muscle of the coronary arteries, and provide an anti-arrhythmic action. For these reasons, studies have to be carried out regarding the inhibition of epoxide hydrolysis in order to render them stable and thus increasing their functionality in cardiovascular disease therapy [42,57,58] (Figure 2).

## 3. Vaso-Protective Activities of n-3 PUFAs

Arterial wall stiffness and endothelial dysfunction indicate a high probability that more or less fatal cardiovascular disease may arise. Overt atherosclerosis is caused by the concomitance of numerous alterations concerning the vessel wall anatomy, vascular endothelium, endothelial-derived factors, and circulating cytokines [35]. It has been shown that n-3 PUFAs have positive effects on numerous molecular and physiological pathways characteristic of the disease as they regulate arterial stiffness and endothelial dysfunction and therefore prevent atherosclerosis and cardiovascular diseases [59,60]. In vivo studies on mice have shown that n-3 PUFAs fatty acids inhibit atherogenesis as they reduce the lipid deposition in the arterial layers, thus inhibiting the Low Density Lipoproteins (LDL) uptake, favouring their removal from the aortic media and inhibiting the lipoprotein lipases synthesis [61].

A diet rich in ω-3 decreases the macrophages and pro-inflammatory markers favouring anti-atherogenic activity [62].

Furthermore, studies conducted in cell lines treated with EPA and DHA have shown a reduction in the proliferation of vascular smooth muscle cells when ω-3 are inserted into membrane phospholipids thereby slowing the progression of cell cycle [63,64]. Similar results were obtained by observing the coronary arteries of subjects who had taken fish oil supplementation. The n-3 PUFAs fatty acids act favourably on plaque stability and prevent breakage thus reducing the onset of thrombotic phenomena [64].

Arterial wall stiffness is determined by active and passive mechanisms of the arterial hemodynamics. Endothelium cells are controlled by molecular and cellular mechanisms which influence the elasticity and mechanical characteristics of the vessels [65].

EPA and DHA act on these mechanisms. A further beneficial effect of n-3 PUFAs intake on wall stiffness is blood pressure reduction. In vivo and clinical studies have documented that a high heart rate, directly related to the progression of arterial stiffness, increases the risk of cardiovascular events. In fact, intake of n-3 PUFAs decreases heart rate and promotes recovery after physical activity. It is also thought that the heart rate reduction is determined by direct effects on cardiac electrophysiology [64]. It is possible that EPA and DHA both modulate the balance between the parasympathetic and sympathetic systems as well as the aortic stiffness and muscle sympathetic nerve activity, thus favouring the neurogenic autonomic function of cardiovascular system [40,66,67].

### 3.1. N-3 PUFAs and Atherosclerosis

In vivo studies show that n-3 PUFAs have the capacity to reverse atherosclerosis, therefore a diet richer in ω-3 protects against coronary artery disease (CAD). It is known that EPA and DHA accumulate in adipose tissue. Thus, in this tissue, they could act on the altered endocrine function, reduce low-grade inflammation caused by obesity, and regulate adipokine gene expression [68]. In patients with stable coronary artery disease, n-3 PUFAs may be added to pharmacological and interventional therapy as they improve adipokine plasma levels. In particular, the plasma levels of adiponectin are enhanced while those of leptin are reduced after 1-month of ω-3 administration. This is especially important in patients with coronary artery disease and many risk factors, because the statins used in pharmacological treatment could influence adipokines plasma levels [22].

It was suggested that in white adipose tissue (WAT) n-3 PUFAs modulates the release of adipokine and cytokines. As key mediators n-3 PUFAs effect on adipose tissue was proposed the protein GPR120 and the receptor PPAR-ϒ [69]. It was proposed that this process involve the suppression of IKK complex activation and JNK phosphorylation, and the reduction of TNF-α secretion, or the formation of a complex between GPR120 and β-arrestin2 down-regulating inflammatory pathway. Further n-3 PUFAs reduce the levels of macrophage, IL-6, TNF-a, MCP-1, and inducible nitric oxide synthase (iNOS) in WAT. Meanwhile, n-3 PUFA PPAR-ϒ-binding promotes the production of anti-inflammatory and insulin-sensitizing adipocytokines [69].

About anti-atherosclerotic effect, n-3 PUFAs and its metabolites, SPMs, could be helpful in counteract atherosclerosis related inflammation [70]. It was proposed that this protective effect occurs through an increase of n-3 PUFAs metabolites (18-monohydroxy EPA, RvE1, RvD1) and a reduction of eicosanoids, such as thromboxane A2 or leukotriene B4, which are pro-inflammatory mediators. Both of these actions determines a reduction in adenosine diphosphate induction of platelet aggregation, coagulation and thrombosis, and a decrease of pro-inflammatory cytokines interleukin-6 (IL-6). Overall, these effects lead to higher plaque stability increasing thicker fibrous cap, reducing oxidized LDL uptake, migration of vascular smooth muscle cells, lesional oxidative stress and necrosis, reducing atherosclerotic lesions progression and vascular inflammation [71]. 

The actions on atherosclerotic plaque was observed in an animal model of hyperlipidemia where pre-treatment with EPA reduced lipid deposition, macrophage adhesion, VCAM-1, ICAM-1 and MCP-1 on endothelial cells and increase content of collagen and smooth muscle cells in atherosclerotic lesions [72].

Further, in a model of diet induced atherosclerosis, in ApoE * 3Leiden transgenic mouse, the endogenous oxidation product of the n-3 PUFA, eicosapentaenoic acid (EPA), RvE1, was administered alone or in combination with atorvastatin for 16 weeks, showing a reduction of atherosclerotic lesions in both type of treatment, more pronounced in co-treatment. Even the pathway of Interferon gamma and TNF-α were downregulated, suggesting an anti-inflammatory action of RvE1 in atherogenesis [73].

The beneficial effect of n-3 PUFAs on atherosclerotic lesions was also due to the modulation of cellular oxidative stress and of total antioxidant status, as have been demonstrated in apolipoprotein E knockout mice. ApoE (−/−) mice fed a diet rich in fish oil for 10 weeks showed an increase of antioxidant enzymes activities such as SOD and catalase (CAT) activities [74]. Fish oil rich diet in ApoE (−/−) mice also increased NO production and eNOS expression and lowered iNOS and reduce lipid peroxidation [75].

Also on vascular endothelial cells n-3 PUFA reduced oxidative stress- induced DNA damage. Under H_2_O_2_ stimulation in aortic endothelial cells, EPA and DHA reduced γ-H2AX foci formation and the activation of kinase ATM, suggesting a reduction of DNA damage response. Further, EPA and DHA also decrease intracellular reactive oxygen species and increase antioxidant defence (heme oxygenase-1, thioredoxin reductase 1, ferritin light chain, ferritin heavy chain and manganese superoxide dismutase) mediating upregulation of NRF2 result in cardiovascular protection [76].

Moreover, in a clinical retrospective analysis of 121 patients conducted in Japan, it was reported that the use of EPA in patients with CAD undergoing percutaneous coronary intervention decreased mean lipid index, macrophage grade and plaque instability [77].

### 3.2. The Effect of n-3 PUFAs in Platelet Function

It is known that n-3 PUFAs can inhibit normal platelet function, thus indicating platelet involvement in EPA and DHA mediated cardioprotection. In fact, numerous studies indicate that platelets treated with EPA and DHA show a reduction in the rate of thrombin formation and the exposure of platelet phosphatidylserine. This treatment reduces thrombus formation and modifies the processing of thrombin precursor proteins [78]. An additional result indicates that when whole blood is treated with ω-3, there is more occlusion time and less fibrin accumulation under flow conditions. Moreover, in vitro studies show that n-3 PUFAs reduce, without eliminating, the procoagulant ability of platelets; this represents one of the cardioprotective mechanisms of ω-3 in subjects with a diet rich in EPA and DHA [79,80].

Along with anti-coagulant action, PUFAs exert anti-platelet effect. Indeed, PUFAs are important constituent of platelet membrane. Free PUFAs were produced by cytoplasmic phospholipase A2 action and they are oxygenated by cyclooxygenase, lipoxygenase and CYP450 into oxylipins. These oxylipins are lipid mediators that regulate platelet function and thrombosus formation. It was proposed that n-3 PUFAs reduced platelet aggregation and thromboxane release through regulation of COX-1 and 12-LOX. In particular, EPA compete with arachidonic acid for platelet COX-1 inhibiting this pathway, reducing thromboxane A2 and increasing the release of other thromboxane, such as TXA3, and suppressing thromboxane receptor. Meanwhile, EPA can increase prostaglandins and NO synthesis in endothelial cells [81,82].

Moreover, it was found that higher levels of platelet phospholipis n-3 PUFA were related to reduced mortality for CVD [83], and that n-3 PUFA supplements is helpful in reduction of platelet aggregation with reduction in ADP-induced platelet aggregation in patients with cardiovascular diseases even if this effect was not proven in healthy patients [84].

It was proposed that anti-platelet effect of n-3 PUFA could depend on gender. EPA seems to have more effectiveness in male while DHA in female as observed by an inverse correlation between testosterone levels and platelet aggregation after EPA administration and the observation of the higher levels of serum DHA in women independently from diet presence of DHA [85].

### 3.3. N-3 PUFAs Index and Coronary Artery Disease (CAD)

The N-3 PUFAs index represents the EPA + DHA percentage contained in the erythrocyte membranes [86]. It has been shown that the fatty acid content in the red blood cell membrane is related to their content in the myocardium. Furthermore, the fatty acids composition of red blood cells is more stable than that of plasma. For this reason, the fatty acid content in the heart can be estimated by measuring the fatty acid content in the erythrocyte membrane. Since it is known that variations in fatty acid content can influence cardiovascular events, it is essential to measure their contents in the erythrocytes membrane in addition to measuring the n-3 PUFAs index. These parameters vary with the diet, therefore they are measured in clinical studies to verify n-3 PUFAs effects as it is known that a higher consumption of n-3 PUFAs causes changes in fatty acids composition in red blood cell membranes, with an increase of the ω-3 index and the lowering of oleic acid. This is very important because this condition reduces the risk of Coronary Artery Disease in the general population [87].

The ratio of serum EPA/arachidonic acid (AA) was considered as a marker of the potential risk of CAD [88]. In a retrospective clinical trials (TREAT-CAD study) of 149 CAD patients who had taken EPA as dietary supplement or not and CAD patients with an EPA/AA ratio > 0.4 taken as cut off, EPA reduced the cumulative incidence of cardiovascular death and improve the long-term prognosis in CAD patients with an EPA/AA ratio ≤ 0.4 [89].

Analysis of serum n-3 PUFAs levels in Chinese in-patients (*n* = 460) with multiple cardiovascular risk factors or an established diagnosis of CAD, concluded that these levels were lower in patients with CAD than those with cardiovascular risk factors and that high serum n-3 PUFAs concentration is associated with decreased CAD proportion at a relatively younger age [90]. 

When taken with an anti-atherosclerotic drug, such as statin, EPA modulate high-density lipoprotein particle size (HDL) in stable CAD patients. In particular, EPA reduced serum levels of HDL3, with an increase of the EPA/AA ratio and of the HDL2/HDL3 ratio. It meant that EPA promoted the conversion of HDL3, with low levels of lipid content, in to large HDL particles that contain more lipid, suggesting a beneficial therapeutic effect of EPA supplementation in statin therapy in counteract coronary artery disease [88].

## 4. The Cardio-Protective Response of n-3 PUFAs Supplementation

As previously described, the n-3 PUFAs have a beneficial role in cardiovascular health; they reduce triglyceride levels enhance high-density lipoprotein levels, and reduce platelet aggregation by preventing coronary arteries occlusion [59,60]. Moreover, they promote a normal heart rhythm, increase arterial compliance, reduce atherosclerosis, and have an anti-inflammatory action [12]. In the Gruppo Italiano per lo Studio della Sopravvivenza nell’Infarto Miocardico (GISSI) trial, the group supplemented with n-3 PUFAs showed a significant reduction in cardiovascular, coronary and sudden cardiac death. The American Heart Association (AHA) recommends the dose of >1 g/day of EPA + DHA dosage defined in the GISSI-Prevenzione study to patients suffering from CAD. This dose lowers triglyceride levels, preserves cardiac function, and decreases the risk of coronary heart disease. In fact, numerous clinical studies report the anti-hypertensive and anti-hyperlipidemic action of n-3 PUFAs [91,92,93,94,95].

They reduce hyperlipidemia because they control gene expression, simultaneously inhibiting lipogenesis and activating lipolysis, and increasing fatty acid β-oxidation. n-3 PUFA therapy has shown promising results in primary and in secondary prevention of cardiovascular diseases [11,42,57,87].

### 4.1. The Effects of n-3 PUFAs in Myocardial Ischemia and Reperfusion

Many studies show that the heart undergoes oxidative stress following myocardial I/R [96,97]. This condition increases ROS synthesis, while at the same time decreasing antioxidant synthesis, thus generating the oxidation of biomolecules and consequent cellular damage [98]. Increased intake of EPA and DHA changes the fatty acid content of myocardial membrane phospholipids providing a higher percentage of DHA and an increase in peroxidisability index values. Nevertheless, there is a reduction in oxidative damage following I/R. In fact, in the I/R condition and greater DHA availability, documentation shows both an increase in myocardial peroxidation with greater basal fatty acid peroxidation and a greater chronic activity of endogenous antioxidant enzyme Mn-SOD together with a decrease in lipid oxidation due to I/R and consequent reduction of heart attack cases. Therefore, during ischemic preconditioning (IPC), ROS have a dual function; generating cellular damage and activating protective signalling processes at the same time. In this context, EPA and DHA intake is linked to both reduction in the ROS levels through lipid peroxidation and increased endogenous antioxidants availability. ROS and endogenous antioxidants are considered, respectively, triggers and mediators of delayed phase ischemic preconditioning. Therefore, the addition of fish oil in the diet is fundamental for the role of n-3 PUFAs in the form of the delayed phase IPC and also because a constant intake of n-3 PUFAs inserted into the membrane phospholipids guarantees constant resistance to damage caused by I/R. For this reason, ischemic protection by n-3 PUFAs is called “nutritional preconditioning” [44,99]. In the heart, n-3 PUFAs act mainly at the mitochondrial level, in fact, they increase the expression of the mitochondrial antioxidant enzyme Mn-SOD; leaving the expression of the other antioxidants such as copper zinc superoxide dismutase (CuZnSOD) and GSx unchanged. Experiments conducted in vivo on mice show that increased Mn-SOD activity resulted in sustained cardioprotection due to heat stress and in the delayed window of ischemic preconditioning. Instead, in the heart, increased expression of Mn-SOD reduces frequency, improves efficiency of oxygen consumption, and ameliorates contractile function. These properties are common to fish oil when taken as a dietary source. In addition, research conducted in rats shows that some fish oil properties, such as lower oxygen consumption in the heart, less sensitivity to I/R damage, and arrhythmias, are due to changes in mitochondrial Ca^2+^. In order for the n-3 PUFAs to carry out their cardioprotective action, they must enter and remain in the myocardium constantly for at least seven days, while their action can last for months, thanks to their lasting insertion in the membrane [28,44].

It has been shown, therefore, that the presence of fish oil in the diet increases the percentage of n-3 PUFAs content in the myocardium membrane, thus increasing the basal peroxidation of cellular fatty acids and consequently increasing Mn-SOD activity. In hearts subjected to acute regional I/R, there is a reduction in stimulated lipid oxidation and myocardial damage. The constant physiological stress generated by this intake, called “oxidative shielding”, if also generated by minimum and regular fish oil doses, could confirm its cardioprotective effect [44]. Beneficial effects result in a reduction in myocardial oxygen consumption and heart rate, with an increase in coronary reserve. Consequently, n-3 PUFAs generates protective preconditioning effects on the damage caused by I/R; thereby improving post-ischemic recovery [28]. In addition to EPA and DHA, alpha linolenic acid (ALA), present in plant foods, can also have cardioprotective effects, particularly during an ischemic attack. This statement derives from data obtained in experiments conducted on cardiomyocytes isolated from adult rats. Cultured cardiomyocytes were pre-treated with ALA for 24 h and subsequently subjected to three different conditions; a group of control cells exposed to non-ischemic conditions, a group exposed to simulated ischemia (ISCH) and another to simulated I/R. It has been shown that in pretreated cardiomyocytes, ALA is incorporated into phosphatidylcholine, thus generating a protective effect on cardiomyocytes subjected to ischemia or I/R. It is likely that ALA has this effect by inhibiting DNA fragmentation during the apoptotic process. Furthermore, ALA inhibits the synthesis of two pro-apoptotic oxidized phosphatidylcholine (OxPC) species that increase significantly during ischemia and I/R, thus resulting in the impairment of apoptosis [7].

### 4.2. The Potential for n-3 PUFAs Supplementation in Heart Failure

In recent years, numerous studies have been conducted concerning the heterogeneous and complex syndrome known as heart failure (HF). In particular, heart failure with reduced ejection fraction (EF) and heart failure with preserved ejection fraction (HFpEF) were defined [100,101,102,103]. The majority of the population is affected by heart failure with preserved ejection fraction, prevalent in the ageing population. Moreover, the common therapies for heart failure are ineffective in cases of heart failure with preserved ejection fraction. Numerous clinical studies document the lower rates of mortality and hospitalization for CHD associated with higher blood levels of n-3 PUFAs fatty acids. Many meta-analyses confirm these data indicating that the risk of CHD and consequently that of sudden death are reduced in the presence of n-3 PUFAs [87,104]. In literature, few studies are reported regarding the effect of n-3 PUFAs on heart failure in vivo and in vitro. Furthermore, no animal model is able to reproduce the phenotypic change and the complex pathophysiology of heart failure with preserved ejection fraction. Therefore, a surgical model of after load induced heart failure, defined as transverse aortic constriction (TAC), has been recreated, which causes some effects of remodelling heart failure with preserved ejection fraction such as: hypertrophy, interstitial cardiac fibrosis, and diastolic dysfunction. In this TAC model, it has been proved that a diet rich in n-3 PUFAs prevents the interstitial fibrosis onset and diastolic heart dysfunction. Furthermore, a direct effect of n-3 PUFAs on cardiac fibroblasts has been reported which prevents the onset of fibrosis. Such evidence suggests that a diet rich in n-3 PUFAs may represent a new therapy for heart failure with preserved ejection fraction [105] Important data indicate that EPA prevents fibrosis by a particular mechanism of action, in fact, it does not accumulate in cardiac myocytes or in fibroblasts. Furthermore, the FFA receptor 4 (FFAR4), a G protein-coupled receptor (GPR) for long chain fatty acids such as EPA and DHA, has proven effective in the prevention of fibrotic signalling in adult cardiac fibroblasts cultures. Studies to define the link between the FFA4 receptor and the n-3 PUFAs were conducted on FFAR4 knockout mice, indicating that in macrophages these fatty acids activate the receptor-mediated anti-inflammatory signalling. Recently, a meta-analysis reported that in the blood of patients treated with a fixed dose of EPA + DHA significantly different levels of these fatty acids were recorded and, moreover, higher values of these were correlated with a lower incidence of CHD risk [8,105,106]. DHA may have a particular cardioprotective role in heart failure conditions caused by pressure overload. These properties are linked to the ease with which DHA is inserted into cardiac fibroblasts and myocytes membranes, modifying their elasticity, fluidity, ion permeability, phase behaviour, fusion, and protein function. Furthermore, DHA inhibits the opening of the mitochondrial transition pores as reported in data obtained in the transverse aortic constriction surgical model. When DHA and EPA are present in the diet together, lower levels of EPA are needed to have the same decrease in fibrosis, indicating that DHA could potentiate the effect of EPA on fibrosis inhibition [105].

## 5. Adverse Effects of PUFAs in Cardiovascular Risk Factors

As mentioned above, n-3 PUFAs are a widely used dietary supplements. However, besides their antioxidant and anti-inflammatory properties, which ascribe to the cardiovascular and vasoprotective actions, n-3 PUFAs are also responsible of side effects [20]. These side effects could be related both to high intake of n-3 PUFAs or to oxidative reactions [63].

Oxidation of n-3 PUFAs could develop following bad storage condition and usage, temperature, light, bad processing conditions, fish cooking, refining process or presence of molecular oxygen responsible of cytotoxicity, genotoxic effects reducing nutritional values of n-3 PUFAs [19,20].

Indeed, n-3 PUFAs is prone to undergo oxidation leading to peroxyl radical formation initiating radical reactions with any hydrogen-donating substance [107].

The progression of lipid peroxidation determines the formation of secondary reactions products, overall leading to the formation of fatty acid peroxides, aldehydes, alcohols, isoprostanes and neuroprostanes [20,107].

It was observed that a dietary oxidized n-3 PUFAs, in particular intestinal absorption of an oxidized n-3 PUFAs end-product such as 4-hydroxy-2-hexenal (4-HHE), induce oxidative stress and inflammation [19]. Mice fed high fat diet, containing lipid mixture of oxidized n-3 PUFAs (EPA and DHA, both in form of triacylglycerols or phospholipids) for eight weeks, showed increased plasma levels of 4-HHE, IL-6 and MCP-1. Further, oxidized n-3 PUFAs, enhance inflammatory response through the activation of NF-kB pathway in small intestine tissue, and enhance levels of glutathione peroxidase and GRP78 as signals of redox stress and an effort to counteract inflammation and oxidative stress [19].

In a single blind clinical trial, 52 women take omega-3 fatty acid supplements in capsules, which were at different levels of oxidation, less oxidized oil pills, highly oxidized oil pills or no capsules, and a rich fish diet for all groups for 30 days. The analysis of total cholesterols, triglycerides and glucose at the beginning and the end of the trial, showed a reduction in triglyceride and cholesterol levels taking less oxidative omega-3 capsules rather than high oxidative omega-3 capsules, which revert the effect on lipid profile. Further, the diastolic and systolic pressure levels decrease in less oxidized n-3 capsules, while no differences were present in highly n-3 oxidized capsules, overall demonstrating the importance of oxidative levels of n-3 supplements [108].

Although beneficial anti-platelet effects [52,53], it was proposed that an high intake of fish oil could increase bleeding time [109,110,111,112,113] and that the consumption of 3-4 g/die of EPA and DHA (helpful in patients with hypertriglyceridemia), moderately increase bleeding times, suggesting that particular attention is needed in patients with anticoagulant drugs therapy [63,113,114,115]. On the other hand, several studies suggest n-3 PUFA are not bleeding risk factor [116,117].

Even if it was proposed that PUFAs affects glucose regulation in obese individuals, probably enhancing hepatic gluconeogenesis, mediating hepatic fatty acid oxidation [118], an increase in fasting glucose and HbA1c after intake of fish or fish oil, was observed in a few study [119].

Further, the high presence of methylmercury, dioxins and polychlorinated biphenyls as contaminants in fish, is debated. Indeed, the high presence of these food pollutants determines adverse effects on CAD [120,121], counteracting cardiovascular benefits of EPA and DHA in accordance on environmental levels. Diets, in which fish represent the major source of contaminants, as observed in several European countries, as well as in Canada and Israel, determined a reduction of the cardio-protective effects of n-3 PUFAs versus myocardial infarction due to the presence of methyl mercury [121,122] or polychlorinated biphenyls [123].

Finally, it was reported that, in patients with chronic heart failure of New York Heart Association class II–IV, the administration of 1 g n-3 PUFA (850–882 mg eicosapentaenoic acid and docosahexaenoic acid as ethyl esters in the average ratio of 1:1·2) determines gastrointestinal adverse effect of minor clinical relevance. However, the rate of patients that discontinued n-3 PUFA because of this side effect were the same in both placebo and n-3 PUFA group [124].

## 6. Conclusions

The use of supplementation with n-3 PUFA has demonstrated, in recent years, to produce both vaso- and cardio-protective responses, which have widely been demonstrated under both in vitro and in vivo settings. These data have been confirmed in patients suggesting their use, along with current therapy, in atherosclerosis prevention, vascular disease management and, finally for heart failure treatment. Many pathophysiological factors seem to contribute on their beneficial effect on vascular system. However, emerging evidence suggest that antioxidant and anti-inflammatory properties of n-3 PUFA seem to play a key role on their efficacy in patients undergoing cardiovascular pathologies (Figure 3). Further studies in larger cohort of patients are required to confirm their extensive use in general population for reducing the risk of severe cardiovascular diseases.

## Figures and Tables

**Figure 1 biomedicines-08-00306-f001:**
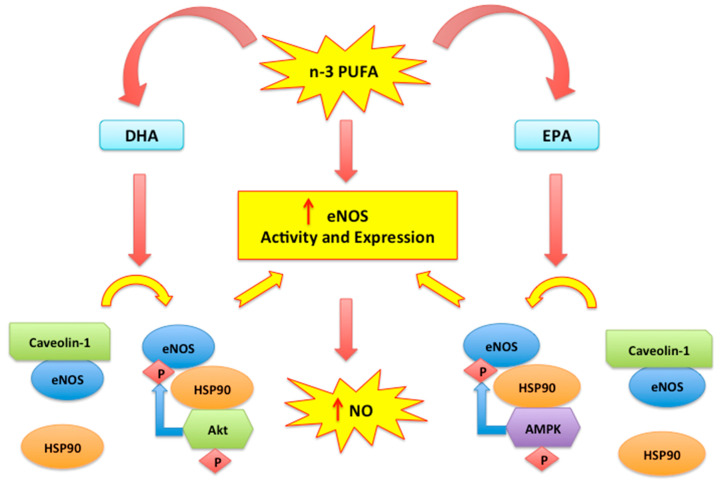
n-3 PUFA mediated eNOS modulation. n-3 PUFA (EPA and DHA) determine a higher NO synthesis by increasing eNOS activity and expression. The unbinding of eNOS/caveolin-1 complex and the recruitment of HSP-90 allow eNOS phosphorylation and activation mediated by Akt and AMPK [18,40,41].

**Figure 2 biomedicines-08-00306-f002:**
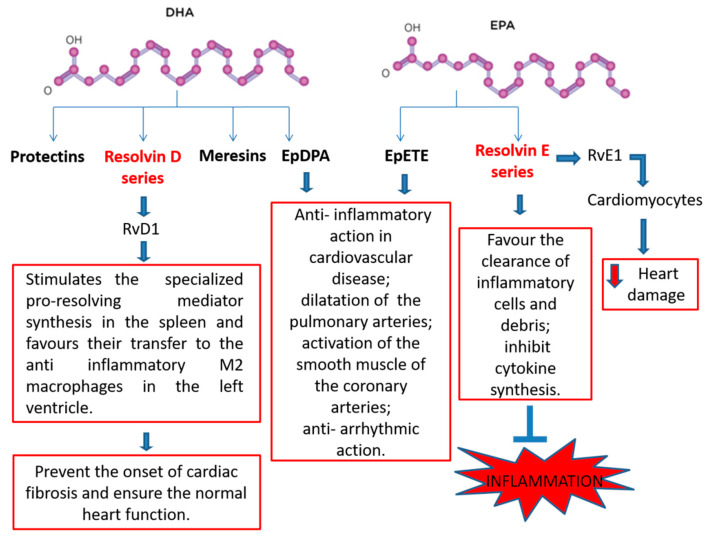
Beneficial effects and potential mechanisms of n-3 PUFA-derived mediators.

**Figure 3 biomedicines-08-00306-f003:**
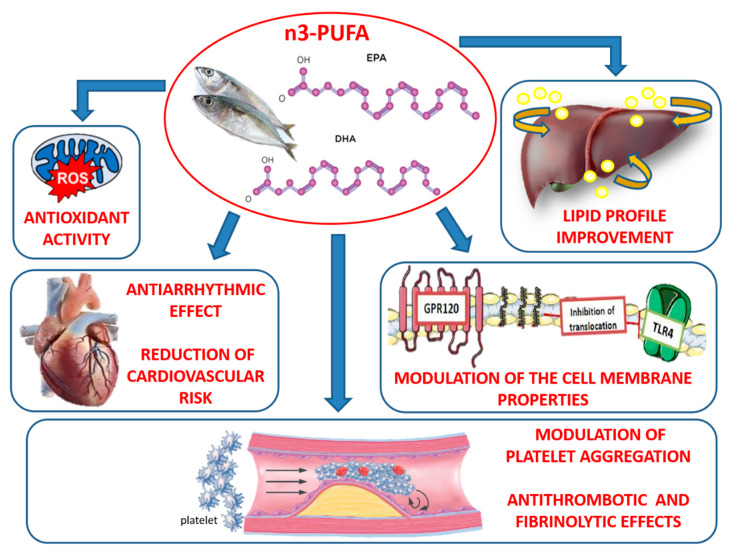
Summary of multifactorial activities of n-3 PUFAs to reduce cardiovascular risk. The n-3 PUFAs (EPA and DHA) improve lipid profile, systemic inflammation and platelet aggregation; furthermore, they have antiarrhythmic, antithrombotic, fibrinolytic effects and antioxidant activity, which concur to reduce heart disease risk.

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
