# Peer review of "The Anti-Inflammatory and Antioxidant Properties of n-3 PUFAs: Their Role in Cardiovascular Protection"

_biomedicines, 2020, doi:10.3390/biomedicines8090306_

Round 1
Reviewer 1 Report
This review introduces antioxidant and anti-inflammatory properties of n-3 PUFAs on cardiovascular protection with focuses on atherosclerosis and cardiac related diseases.
- Lines 53-56, the authors state “In addition to the collected evidence demonstrating the beneficial responses of dietary n-3 PUFAs supplementation on the cardiovascular system, some concerns have been expressed about their potential detrimental effects in different tissues due to their potential pro-oxidant effect [17]. In particular, oxidative stress and n-3 PUFAs are strongly correlated.” The authors only provide beneficial effects but not potential adverse effects in Sections 3 and 4. It is important to discuss the literature in Sections 3 and 4.
- Section 3 is over brief, but Section 4 is more detailed. Is it because more literature evidence for heart disease than for atherosclerosis? Many statements should include references as suggested below (but not limited to the suggestions below):
Line 141: This damage is likely to be caused by reduced synthesis of SPMs…… references for this statement?
Lines 144-146: In fact, rats treated with RvE1 after ischemia/reperfusion suffer from minor infarct. In particular, the results demonstrate that RvE1 reduces heart damage by direct action on cardiomyocytes. References for these statements? “minor infarct” should be changed to a phrase that compares treatment with or without RvE1 such as “less infarcted area” or “less severity of myocardial infarction”.
Lines 164-168: …they reduce the lipid deposition in the arterial layers, thus inhibiting the Low Density Lipoproteins (LDL) uptake, favoring their removal from the aortic media and inhibiting the lipoprotein lipases synthesis. A diet rich in w-3 decreases the macrophages and pro-inflammatory markers favouring anti-atherogenic activity. - References to support these statements?
Lines 168- 180. The statements need References
Section 3.1 and 3.2 – Limited references
Lines 299-302: In this TAC model, it has been proved that a diet rich in n-3 PUFAs prevents the interstitial fibrosis onset and diastolic heart dysfunction. Furthermore, a direct effect of n-3 PUFAs on cardiac fibroblasts has been reported which prevents the onset of fibrosis. – Please provide references.
The authors used many times of “numerous… studies” or “many studies” with no reference or just one reference to support the statements.
- Lines 73-75: “A diet rich in n-3 PUFAs has a positive effect on blood pressure, heart rate, left ventricular diastolic filling, and endothelial functions [22].” – Does “positive” mean “beneficial”? Same comments on the “positive” in Line 218.
- Lines 79-80: “In mitochondria, DHA acts on many pathways; reducing oxidative stress and c oxidase activity as well as increasing Mn-SOD activity.” - What does “c oxidase” mean?
- Figure 1. title “ eNOS modulation n-3 PUFA mediated”. Suggest to change to “n-3 PUFA mediated eNOS modulation.” Also suggest to include appropriate references that support these pathways in the legend.
- Lines 139 and the paragraph starts with Line 147: Suggest to describe RvD1 together, rather than in two paragraphs with no transition.
- Line 132: 2.4. N-3 PUFA-derived mediators: a schematic graph to show N-3 PUFA-derived mediators, their beneficial effects, and potential mechanisms would help readers understand these components better.
- Title: “antinflammatory” is a typo?
- Lines 129-131: Moreover, n-3 PUFAs regulate leukotrienes, prostaglandins and thromboxanes synthesis, since, through activation of cytosolic phospholipase A2 (cPLA2), cyclooxygenase 2, and production of 131 PGE2, as a cPLA2 inhibitor, modify the metabolic pathway of arachidonic acid [15, 16, 37, 38]. – Should “since” be removed?
- Line 51: regulate the COX pathway, and promote 51 proinflammatory PGE2 synthesis [15] – add “.” after “[15]”.
Author Response
Point-by-point response to the reviewers:
Reviewer 1
This review introduces antioxidant and anti-inflammatory properties of n-3 PUFAs on cardiovascular protection with focuses on atherosclerosis and cardiac related diseases.
- Lines 53-56, the authors state “In addition to the collected evidence demonstrating the beneficial responses of dietary n-3 PUFAs supplementation on the cardiovascular system, some concerns have been expressed about their potential detrimental effects in different tissues due to their potential pro-oxidant effect [17]. In particular, oxidative stress and n-3 PUFAs are strongly correlated.” The authors only provide beneficial effects but not potential adverse effects in Sections 3 and 4. It is important to discuss the literature in Sections 3 and 4.
A.1. We thanks the reviewer for the suggestion. A section 5 titled “Adverse effects of PUFAs in cardiovascular risk factors” , had been included to discuss the literature.
2.Section 3 is over brief, but Section 4 is more detailed. Is it because more literature evidence for heart disease than for atherosclerosis? Many statements should include references as suggested below (but not limited to the suggestions below):
A.2. Section 3 provides now more detailed informations and new references have been added.
Line 141: This damage is likely to be caused by reduced synthesis of SPMs…… references for this statement?
We thanks the reviewer for the suggestion. Reference had been added.
Lines 144-146: In fact, rats treated with RvE1 after ischemia/reperfusion suffer from minor infarct. In particular, the results demonstrate that RvE1 reduces heart damage by direct action on cardiomyocytes. References for these statements? “minor infarct” should be changed to a phrase that compares treatment with or without RvE1 such as “less infarcted area” or “less severity of myocardial infarction”.
We thanks the reviewer for the suggestions. The reference had been provided, and the sentence had been changed.
Lines 164-168: …they reduce the lipid deposition in the arterial layers, thus inhibiting the Low Density Lipoproteins (LDL) uptake, favoring their removal from the aortic media and inhibiting the lipoprotein lipases synthesis. A diet rich in w-3 decreases the macrophages and pro-inflammatory markers favouring anti-atherogenic activity. - References to support these statements?
More references have been provided.
Lines 168- 180. The statements need References
We thanks the reviewer for the suggestion. References have been provided.
Section 3.1 and 3.2 – Limited references
More reference had been provided.
Lines 299-302: In this TAC model, it has been proved that a diet rich in n-3 PUFAs prevents the interstitial fibrosis onset and diastolic heart dysfunction. Furthermore, a direct effect of n-3 PUFAs on cardiac fibroblasts has been reported which prevents the onset of fibrosis. – Please provide references.
We tanks the reviewer for the suggestion. Reference had been provided.
The authors used many times of “numerous… studies” or “many studies” with no reference or just one reference to support the statements.
We thanks the reviewer for the suggestion. More references have been provided.
- Lines 73-75: “A diet rich in n-3 PUFAs has a positive effect on blood pressure, heart rate, left ventricular diastolic filling, and endothelial functions [22].” – Does “positive” mean “beneficial”? Same comments on the “positive” in Line 218.
We tanks the reviewer for the suggestion. The sentence had been changed to “A diet rich in n-3 PUFAs has a “beneficial” effect on blood pressure, heart rate, left ventricular diastolic filling, and endothelial functions” and to
“As previously described, the n-3 PUFAs have a “beneficial” role in cardiovascular health; they reduce triglyceride levels enhance high-density lipoprotein levels, and reduce platelet aggregation by preventing coronary arteries occlusion”
- Lines 79-80: “In mitochondria, DHA acts on many pathways; reducing oxidative stress and c oxidase activity as well as increasing Mn-SOD activity.” - What does “c oxidase” mean?
We thanks the reviewer fot the suggestion. The sentence had been changed to “In mitochondria, DHA acts on many pathways; reducing oxidative stress and cytochrome c oxidase (complex IV) activity as well as increasing Mn-SOD activity.”
- Figure 1. title “ eNOS modulation n-3 PUFA mediated”. Suggest to change to “n-3 PUFA mediated eNOS modulation.” Also suggest to include appropriate references that support these pathways in the legend.
We thanks the reviewer for the suggestion. Figure 1 title had been changed to “n-3 PUFA mediated eNOS modulation.” And references to support these pathway have been included.
- Lines 139 and the paragraph starts with Line 147: Suggest to describe RvD1 together, rather than in two paragraphs with no transition.
We thanks the reviewer for the suggestion. The description of RvD1 had been grouped.
- Line 132: 2.4. N-3 PUFA-derived mediators: a schematic graph to show N-3 PUFA-derived mediators, their beneficial effects, and potential mechanisms would help readers understand these components better.
We thank the reviewer for the suggestion. A schematic figure named “fig.2” has been provided.
- Title: “antinflammatory” is a typo?
We thanks the reviewer for the suggeston. The word “antinflammatory” had been changed to” anti-inflammatory”
- Lines 129-131: Moreover, n-3 PUFAs regulate leukotrienes, prostaglandins and thromboxanes synthesis, since, through activation of cytosolic phospholipase A2 (cPLA2), cyclooxygenase 2, and production of 131 PGE2, as a cPLA2 inhibitor, modify the metabolic pathway of arachidonic acid [15, 16, 37, 38]. – Should “since” be removed?
We thanks the reviewer for the suggestion. Sentence had been modified as suggested; “since” had been removed
- Line 51: regulate the COX pathway, and promote 51 proinflammatory PGE2 synthesis [15] – add “.” after “[15]”.
We thanks the reviewer for the suggestion. The “.” Had been added.
Reviewer 2 Report
In this review paper, the authors summarized the antioxidant and anti-inflammatory properties of polyunsaturated fatty acids and their effect of veso- and cardio-protection and on heart failure. The reviewer considers that this manuscript provide a topic of interest to the audiences in this field with recent literature report. However, this topic was well documented recently by others and the information provided in this manuscript did not provide any new insights on the properties and effects of polyunsaturated fatty acids. The information summarized in the content were not well organized and some were merely a description without insightful input from the authors. The value of this reviewer paper, therefore, is very limited. In addition, the authors used many abbreviations in the content. However, they were not used properly. Some abbreviations were used before the original terms were presented. Other abbreviations were presented for the first time but not used in the following content. The English in Figure 1 legend needs to be corrected. The illustration in Figure 2 needs to be presented more logically. Besides, grammar errors and typos reduced its readability.
Author Response
Reviewer 2
In this review paper, the authors summarized the antioxidant and anti-inflammatory properties of polyunsaturated fatty acids and their effect of veso- and cardio-protection and on heart failure. The reviewer considers that this manuscript provide a topic of interest to the audiences in this field with recent literature report. However, this topic was well documented recently by others and the information provided in this manuscript did not provide any new insights on the properties and effects of polyunsaturated fatty acids. The information summarized in the content were not well organized and some were merely a description without insightful input from the authors. The value of this reviewer paper, therefore, is very limited. In addition, the authors used many abbreviations in the content. However, they were not used properly. Some abbreviations were used before the original terms were presented. Other abbreviations were presented for the first time but not used in the following content. The English in Figure 1 legend needs to be corrected. The illustration in Figure 2 needs to be presented more logically. Besides, grammar errors and typos reduced its readability.
We are aware that many review papers have been written in the last years on the topic of antioxidant and anti-inflammatory properties of polyunsaturated fatty acids and their effect of vaso- and cardio-protection. Nevertheless, in this revised version, according to the suggestions and comments of the reviewers, more detailed literature has been provided. Furthermore, we added a new paragraph related to the potential side effects of PUFA, which is an important aspect (often missing )in cardiovascular risk factors.
Accordingly to the suggestions, the original terms is now presented before the abbreviations. The English in Figure 1 legend has been corrected. The illustration in Figure 2 is presented now more logically. Manuscript has been now edited and proofread by an native English speaker.
Round 2
Reviewer 1 Report
No further comments
Author Response
We thank the Reviewer.
Reviewer 2 Report
In the second version of this review paper, the authors made significant changes of the content with addition of more relevant references, added a new figure and modified a figure for better illustration purpose. The reviewer considers that all the changes have contributed positively to this manuscript which is in a better position for publication in Biomedicines given the following issues raised to be addressed first.
- This manuscript needs a total checkup to make sure all abbreviations are used properly. For example, in line 89 page 2, IR was used as abbreviation without having “ischemia reperfusion” first. In addition, some abbreviation was initiated in the later part of the paper such as IR and ROS in line 319 page 8. This issue possibly happened when many authors contributed to their own parts without checking other parts.
- In figure 3, something is missing following the first arrow on the left.
Author Response
Reviewer 2
In the second version of this review paper, the authors made significant changes of the content with addition of more relevant references, added a new figure and modified a figure for better illustration purpose. The reviewer considers that all the changes have contributed positively to this manuscript which is in a better position for publication in Biomedicines given the following issues raised to be addressed first.
- This manuscript needs a total checkup to make sure all abbreviations are used properly. For example, in line 89 page 2, IR was used as abbreviation without having “ischemia reperfusion” first. In addition, some abbreviation was initiated in the later part of the paper such as IR and ROS in line 319 page 8. This issue possibly happened when many authors contributed to their own parts without checking other parts.
A.1. We thank the Reviewer for the observations. We made a total checkup of the manuscript and now all the abbreviations are used properly. Changes to the manuscript are highlighted, through the "Track Changes" function.
- In figure 3, something is missing following the first arrow on the left.
A.2. We thank the Reviewer for noting the missing part of the figure. Figure 3 has been updated and corrected.